# Aging in Ocular Blood Vessels: Molecular Insights and the Role of Oxidative Stress

**DOI:** 10.3390/biomedicines12040817

**Published:** 2024-04-08

**Authors:** Xiuting Cui, Francesco Buonfiglio, Norbert Pfeiffer, Adrian Gericke

**Affiliations:** Department of Ophthalmology, University Medical Center, Johannes Gutenberg University Mainz, Langenbeckstrasse 1, 55131 Mainz, Germany; fbuonfig@uni-mainz.de (F.B.); norbert.pfeiffer@unimedizin-mainz.de (N.P.)

**Keywords:** aging, blood vessels, vascular aging, pathophysiology, retina, choroid

## Abstract

Acknowledged as a significant pathogenetic driver for numerous diseases, aging has become a focal point in addressing the profound changes associated with increasing human life expectancy, posing a critical concern for global public health. Emerging evidence suggests that factors influencing vascular aging extend their impact to choroidal and retinal blood vessels. The objective of this work is to provide a comprehensive overview of the impact of vascular aging on ocular blood vessels and related diseases. Additionally, this study aims to illuminate molecular insights contributing to vascular cell aging, with a particular emphasis on the choroid and retina. Moreover, innovative molecular targets operating within the domain of ocular vascular aging are presented and discussed.

## 1. Introduction

Aging stands out as a predominant pathogenetic factor intricately influencing the pathophysiology of numerous chronic disorders. Significantly, the clustering of multiple chronic diseases in the elderly diverges from the anticipated distribution based on the composite probability of individual ailments. Aging exerts a multifaceted impact on the development of senescent phenotypes, encompassing the following four key aspects: (1) alterations in body composition; (2) an imbalance between energy supply and demand; (3) dysregulation of signaling networks maintaining homeostasis; and (4) neurodegeneration with impaired neuroplasticity [1]. Moreover, the intricate connection between aging and immune-related diseases underscores the substantial contribution of immune system aging to compromised tissue repair, heightened infection susceptibility, and increased cancer rates. Autoimmune diseases frequently manifest in older age, leading to varying degrees of damage to the body [2,3,4].

In the context of the ocular vascular network, aging induces structural and functional damage, elevating the risk of visual impairment [5]. The impact of aging on the retina and choroid is particularly pronounced, with retinal aging, especially in the macular area of the central retina, posing a significant clinical concern. Age-related macular degeneration (AMD) stands as a leading cause of blindness among older adults, with an expected increase in patients to 288 million by 2040 [6]. Furthermore, advanced age emerges as a major risk factor for retinal vascular occlusion and immunological diseases that may affect the ocular vasculature, such as giant cell arteritis [3].

Recognizing the pivotal role of vascular aging in various ocular disorders and its pathophysiological significance potentially contributing to functional impairment of the choroid and retina, there is a keen interest in exploring the primary molecular pathways involved in these events. This review endeavors to provide a comprehensive exploration of choroidal and retinal vascular aging, delving into the molecular cascades responsible for these processes. Additionally, innovative molecular targets with the potential to counteract this pathophysiological driver are highlighted.

## 2. The Concepts of Aging and Vascular Aging

In the following sections, we delve into fundamental concepts surrounding the cellular mechanisms of aging and senescence, intricately linked with the perturbation of physiological immune functions, thereby giving rise to inflammatory responses. Furthermore, we elucidate particular phenomena manifesting at the vascular level as a result of aging processes.

### 2.1. General Insights into Aging

In the 1960s, Hayflick et al. conducted human cell culture experiments and made a groundbreaking discovery: the inherent limitation of human cells to undergo unlimited divisions [7,8]. This phenomenon, now acknowledged as cellular senescence, denotes the irreversible halting of the cell cycle due to diverse processes, influencing cellular function and disturbing normal tissue homeostasis [9]. Cellular senescence stands as a pivotal contributor to the aging process within the body, with the term “senescence” deriving from the Latin word “senex”, signifying “to grow old” [10]. Aging is a universal process that causes the body’s functions to decline over time [11]. It involves characteristics such as irreversible growth arrest, functional decline, and significant pro-inflammatory changes in the secretome [12]. Throughout the aging process, cellular senescence is exacerbated by various endogenous and exogenous stressors [13,14]. A clinical study led by Coppé and colleagues has elucidated three primary characteristics frequently observed in aging cells: stagnation of cell proliferation, resistance to apoptosis, and the emergence of complex senescence-related secretory phenotypes [15]. Additionally, cellular senescence is characterized by age-related manifestations and can be experimentally accentuated to accelerate aging. Understanding and potentially modifying these features are crucial for therapeutic interventions aimed at slowing, stopping, or reversing the aging process [16]. Dedicated investigations have identified twelve key aging-related features, including genomic instability, dysregulated nutrient sensing, telomere attrition, epigenetic changes, loss of proteostasis, dysfunction in macroautophagy, cellular senescence, dysbiosis, stem cell exhaustion, altered intercellular communication, chronic inflammation, and mitochondrial dysfunction [17]. Inflammatory aging was first named by Franceschi et al. [18]. During the aging process, age-related changes in immune system function are accompanied by chronic sterile inflammation, a mechanism known as “inflammatory senescence” [19]. Inflammatory senescence is due to the widespread dysregulation of multiple inflammatory molecules and innate and adaptive immunity. It occurs due to functional cell defects. An important source of inflammatory signaling in aging organisms is thought to be the accumulation of senescent cells in tissues. Inflammation is one of the determinants of aging and plays an extremely important role in most age-related chronic diseases [20]. Importantly, these characteristics are not isolated but interconnected, exhibiting interactions with each other [21]. Other critical agents, such as spatial organization, maintenance of homeostasis, and effective stress responses, have also been recognized as interconnected with these aging-related characteristics [16].

### 2.2. Insights into Vascular Aging

In the 17th century, Thomas Sydenham argued that “A man is only as old as his arteries”, recognizing that the aging process in the body results from the interplay of genetics and external factors [22]. This insight forms the foundation for a contemporary understanding of vascular aging, where the disparity between the body’s chronological age and its vascular age determines the extent of blood vessel aging [23]. Primarily, the changes induced by this condition are most conspicuous in the media of large arteries, leading to significant arterial stiffness, lumen dilation, and wall thickening [24]. Recent insights have expanded our comprehension of age-related microvascular disease, highlighting its importance. Vascular aging can inflict structural and functional impairments in the microcirculation process, impacting crucial functions such as tissue oxygenation, nutrient transport, and waste removal [25]. The cumulative effects of these processes contribute to eventual organ dysfunction. Furthermore, vascular aging influences microvascular perfusion and pathological structural remodeling [26]. Notably, these alterations in phenotypes of endothelial cells, smooth muscle cells, and pericytes contribute to dysfunctional changes in the immune and endocrine systems, compromising their activities [27].

Throughout the aging process, arteries undergo structural and functional deterioration, with implications not only for large vessels like the aorta but also for the microvascular circulation [28]. Vascular aging has emerged as a significant factor closely linked to age-related dementia, including Alzheimer’s disease [29]. Research has also demonstrated the clinical relevance of vascular endothelial cell aging to pulmonary hypertension [30]. Furthermore, various cerebrovascular diseases exhibit close associations with vascular aging. Cerebral small vessel disease (SVD), characterized by alterations in leptomeningeal and brain parenchyma microcirculation, is notably influenced by the aging of brain microvessels, which stands as a critical factor in prognosis [31]. Apart from cerebrovascular diseases, microvascular aging is intricately connected to the pathogenesis of a spectrum of age-related diseases throughout the human body [27].

## 3. Vascular Aging in Ocular Blood Vessels

In the subsequent sections, we delve into the nuanced processes of vascular aging within the eye. Specifically, our focus is on the physiological, pathophysiological, and molecular events affecting the choroid and retina. Furthermore, we shed light on the intricate relationship between vascular aging and age-related ocular disorders.

### 3.1. The Choroid

#### 3.1.1. Choroidal Vascular Circulation: Physiological Significance

The choroid, situated between the retina and the sclera, emerges as a critical tissue primarily composed of pigments and blood vessels. Characterized by its dark brown color, the choroid plays a pivotal role in the human eyeball by supplying essential nutrients and oxygen to the retina [32]. Unique among vascular systems in the body, the choroid acts as a distinct vascular layer within the eye, positioned between the sclera and Bruch’s membrane. Notably, the arteries and veins in the choroid follow a non-parallel course [33].

The choroid constitutes a highly vascularized structure outside the retina, organized into three layers. The outermost layer, Haller’s layer, accommodates the largest blood vessels, while the middle Sattler’s layer contains medium-diameter blood vessels [33]. Adjacent to the inner side of Bruch’s membrane lies the choriocapillaris, comprising fenestrated capillaries [34]. Capillaries branch off from arterioles at nearly right angles, with voluminous and short-diameter arterioles in the macular area. Blood vessels without fenestrations are distributed in Haller’s and Sattler’s layers. The innermost layer, choriocapillaris, situated near and within the layer of Saler, features fenestrated vessels facilitating the supply of oxygen and nutrients to the outer retina [35]. Endothelial cells in the choriocapillaris lack tight junctions but possess multiple pores. Blood supply to the iris and ciliary body is derived from the long posterior ciliary artery and anterior ciliary artery, both branching from the ophthalmic artery—the initial branch of the internal carotid artery [36]. Venous drainage occurs through vortex veins and primarily through the anterior ciliary veins of the ciliary body [37]. The short posterior ciliary artery, branching from the ophthalmic artery, penetrates the sclera, forming fan-shaped lobules to provide local blood supply [38]. These arteries create a three-layered choroidal microcirculatory system by piercing the sclera [39].

As a vital source of nutrition for the outer retina, the choroid stands out with one of the highest blood flows and metabolic demands in the body [40,41]. It plays a dynamic role in adjusting the position of the retina by altering its thickness and releasing growth factors involved in vascularization, scleral remodeling, and eye growth regulation. Additionally, the choroidal circulation possesses an automatic adjustment function in response to changes in perfusion pressure [42]. This adaptation ensures eye protection during temperature fluctuations and enhances oxygen supply to the retina when retinal circulation is insufficient [43].

Figure 1 offers an overview of the retinal layers and vasculature as well as the structure of the choroid, comprising a layer of choriocapillaris along the Bruch’s membrane.

#### 3.1.2. Choroidal Vascular Aging: Pathophysiological Insights

Clinical data derived from optical coherence tomography (OCT) unveil an inverse correlation between age and choroidal volume, supported by histopathological studies indicating a decline in choriocapillaris density with advancing age [44]. In the context of age-related macular degeneration (AMD), choriocapillaris density exhibits a nuanced trajectory, initially increasing in early stages but eventually undergoing a more pronounced decline compared with normal aging [45]. Microvascular aging is closely associated with Alzheimer’s disease, heart failure, and other age-related conditions [46]. In the context of the eye, the choroid and choriocapillaris undergo morphological and functional changes with aging. Notably, AMD, a leading cause of vision loss among the elderly, shares common risk factors with atherosclerosis [47]. Research underscores the significant role of vascular aging in AMD pathogenesis, involving fundamental mechanisms of cellular and molecular aging such as mitochondrial dysfunction, oxidative stress, and compromised choriocapillaris resistance to molecular stressors [47]. The complex pathophysiology of AMD encompasses oxidative stress, lipid peroxidation, complement dysregulation, and choroidal hypoperfusion [48].

To explore choroidal vascular changes in AMD patients, real-time ocular imaging was employed to assess dynamic vascular alterations after oral administration of sildenafil citrate—a potent vasodilator and phosphodiesterase type 5 (PDE5) inhibitor [49]. The results underscored stromal expansion as a primary contributor to choroidal thickening, with sildenafil having a limited effect on retinal vessel density [50]. Notably, the degree of choroidal vascular response to sildenafil exhibited an inverse trend with age [50]. Vascular aging, an irreversible pathophysiological process, is considered a common feature in age-related cardiovascular and microvascular diseases, including hypertension and retinal vascular disease [46]. Diabetic retinopathy (DR), the most prevalent retinal vascular disease, showcases microvascular abnormalities in the retina and choroid [51]. Experimental studies emphasize the role of choroidal pathology in diabetic patients as a potential contributor to DR [52,53]. Pathological changes in choroidal vessels, akin to those observed in retinal vessels, may be fundamental to DR, with choroidal thickness variations indicating underlying eye diseases [54]. Aging, a major risk factor for fibrotic processes, contributes to organ fibrosis incidence, particularly evident in proliferative diabetic retinopathy and neovascular age-related macular degeneration (nAMD). Fibrosis-related conditions including nAMD and choroidal neovascularization (CNV), secondary to other eye diseases, with aging promoting CNV-induced subretinal fibrosis [55]. Human choroidal vasculature undergoes aging changes, evidenced by age-related decreases in overall choroidal thickness, choriocapillaris density, vessel diameter, and choroidal blood flow. Age-related alterations in choroidal vascular aging involve increased expression of chemokines and complement genes, elevated choroidal macrophage density, and changes in macrophage polarization [56]. In mice, the aging choroid exhibits vascular alterations similar to those observed in humans, characterized by attenuated and tortuous choroidal arteries and reduced choriocapillaris density [57].

Choriocapillaris density significantly decreases with age, and age-related changes in the choroidal vasculature correlate with extensive Bruch’s membrane (BrM) thickness changes, basal linear and basal lamellar deposition, and RPE alterations [58,59]. Two main clinical theories regarding AMD propose that primary damage occurs at the RPE level, leading to secondary changes in the choroidal vasculature, or that impaired choroidal perfusion directly causes RPE dysfunction [60]. Experiments have shown that choroidal volume and blood flow are reduced by approximately 29% in the elderly group compared with young patients, supporting the notion that choroidal changes may contribute to AMD pathogenesis [61]. Normal aging induces spatially specific alterations in choroidal vascular structure, with the choroidal vascular index (CVI) undergoing significant changes as part of the aging process [62]. Other studies also corroborate the concept of age-related loss of choroidal vascular structure, evident in subfoveal choroidal thinning, choriocapillaris thinning, reduced choroidal volume, reduced choroidal vessel density, and an inverse correlation between choroidal vessel diameter and age [56].

Figure 2 illustrates the main pathophysiological players contributing to choroidal vascular aging. 

#### 3.1.3. Molecular Insights into Choroidal Vascular Aging

Common mechanisms underpinning vascular aging encompass oxidative stress, inflammation, impaired protein homeostasis, mitochondrial dysfunction/DNA damage, dysregulated nutrient sensing, and cellular senescence [63]. The complex pathogenesis of AMD, though subject to ongoing debate, unequivocally involves aging and the choroid. Molecular alterations targeting the choroid in AMD are characterized by multifaceted inflammatory mechanisms.

Oxidative stress, a pivotal pathogenic stimulus in neovascular AMD and geographic atrophy, is induced by hypoxia and chemical mediators, leading to increased VEGF production and playing a role in choroidal neovascularization [64,65]. The Nrf2 antioxidant pathway emerges as a crucial homeostatic mechanism against oxidative stress, with studies highlighting its significance in the RPE antioxidant response [47].

Expression of human leukocyte antigen (HLA) in healthy and AMD eyes, primarily on choriocapillaris endothelial cells, is associated with AMD progression [66].

The complement system, normally regulated by complement regulatory proteins, involves the complement factor H (CFH) gene as a key protein in AMD pathogenesis, regulating the alternative pathway through C3b inactivation [67,68]. Mast cells and macrophages are the major leukocytes detectable in the choroid and have been shown to be involved in the pathogenesis of AMD [68]. Choriocapillaris endothelial cells heavily depend on VEGF, which is crucial for their development and maintenance [60].

The results of the study by Steinle et al. indicated that during normal aging, angiopoietin 1 (Ang-1) receptor Tie-2 levels in the choroid/RPE complex increase, pigment epithelium-derived factor (PEDF), which is an angiogenesis inhibitor produced by RPE cells, and VEGF protein levels decrease, while Ang-1 levels remain almost unchanged [69]. Normal aging reduces choroidal blood flow [69]. Another age-related change occurring within the BrM that may contribute to AMD is the accumulation of advanced glycation end products (AGEs) [68]. AGEs are proteins or lipids covalently bound to sugar molecules. In contrast to physiological glycosylation that occurs within cells in the endoplasmic reticulum and Golgi apparatus, AGE formation occurs nonenzymatically and extracellularly [70]. AGEs preferentially accumulate in the extracellular matrix (e.g., the BrM) and may disrupt normal protein function.

Interestingly, in eyes affected by AMD, a contrasting trend between C-reactive protein (CRP) and CFH levels has been observed. CRP levels were notably higher in the BrM and choroidal stroma of early and wet AMD eyes compared with controls, while CFH levels were lower in the BrM and choroids of AMD eyes compared with controls. Additionally, both CRP and CFH levels were significantly reduced in the BrM of atrophic areas in geographic atrophy [71]. CRP monomers (mCRP) are the only form present in choriocapillaris [68]. In the choroid, aging is associated with the pro-inflammatory upregulation of the CCL2-CCR2 axis, which is one of the major chemokine signaling pathways, and genetic ablation of CCL2 reduces age-related inflammatory changes in the choroid, reducing the recruitment of pro-inflammatory myeloid cells and attenuating CNV [72].

Research on aging choroidal endothelial cells in rhesus monkeys reveals high levels of SA-β-Gal and abnormal cytoskeletal activity, sensitizing choriocapillaris to MAC-induced endothelial dysfunction [73]. Increasing evidence suggests that mitochondrial dysfunction plays a critical role in microvascular aging, and the accumulation of DNA damage in the aging vasculature may contribute to replicative senescence [27,74,75,76]. Mitochondrial damage also occurs in AMD, and changes in RPE mitochondrial structure are common in AMD eyes. This mitochondrial damage/mtDNA damage is thought to be an important factor in RPE death in AMD [47]. Figure 3 summarizes the connection between choroidal aging and the development of AMD, as extensively described in the literature [68].

### 3.2. Retina

#### 3.2.1. Retinal Vascular Circulation: Physiological Significance

The retina, originating from the diencephalon during embryonic development, shares characteristics with the central nervous system and the cerebrovascular bed [77]. The central retinal artery and vein branch in the optic nerve head area supply blood to each quadrant of the retina [78]. Unlike the microvascular bed of the peripheral circulation, the retinal microcirculation forms a terminal arterial system devoid of anastomoses and capillary sphincters [79]. Research has found that branching and spatial characterization are key properties of retinal circulation [80]. Blood flow distribution in the retina varies, with temporal retinal vessels being 25% larger than those in the nasal region and exhibiting a higher metabolic rate. Consequently, blood flow is 2–3 times greater in the temporal retina than in the nasal retina [81]. The oxygen consumption of retinal tissue surpasses that of most other tissues in the human body [82].

Retinal circulation plays an important role in oxygen supply and is involved in maintaining retinal homeostasis [78]. Because of continuous conversion of light into neuronal signals, a highly metabolic activity, the retina exhibits a heightened demand for oxygen [77]. The dense vascular system influences the transmission of light signals, prompting the evolution of distinct methods for providing nutrients and oxygen to different regions of the retina. Specifically, the choroidal microcirculation, branching from the posterior ciliary artery, caters to the outer one-third to one-half of the retina [35]. In contrast, the retinal microcirculation, branching off from the central retinal artery, supplies the inner half to two-thirds of the retina [41,77]. Originating from the internal carotid artery, the ophthalmic artery further branches into the central retinal artery, short ciliary artery, long ciliary artery, and anterior ciliary artery.

Being an extension of the central nervous system, the retina’s vasculature mirrors the characteristics of the cerebral vasculature. The blood–retinal barrier, akin to the blood–brain barrier, regulates the reach of circulating blood components to surrounding tissues [83]. Damage to the blood–retinal barrier can lead to various retinal degenerative diseases and blindness [41]. The retina is especially vulnerable to oxidative stress, particularly in the macula, leading to impaired central vision. Therefore, maintaining a regulated microenvironment within the retina, separated from the systemic circulation by the blood–retinal barrier, is crucial [84].

#### 3.2.2. Retinal Vascular Aging: Pathophysiology

The retina, being crucial for vision, relies on a complex blood vessel network. In diabetic retinopathy, retinal microvessels undergo abnormal aging and regeneration, significantly impairing normal retinal function [85]. It has been found that vascular endothelial cells in diabetic retinopathy patients are involved in molecular pathways related to cell aging [86].

Hyperglycemia, characteristic in diabetic patients, induces vascular endothelial dysfunction through various mechanisms such as glycocalyx damage, inflammation, oxidative stress, and cellular senescence activation [87]. The presence of high glucose concentration and AGEs accelerates endothelial cell senescence, a key pathological mechanism in diabetic retinopathy [88]. Long-term exposure to high glucose environments induces retinal endothelial cell aging, accompanied by oxidative stress reactions, increased β-galactosidase activity, p53 expression, and elevated secretion of factors like IL-6 and VEGF, characteristic of diabetic retinopathy [88,89,90].

The aging process of the human retina often triggers oxidative stress reactions, inducing significant alterations within the retina [91]. Nag et al. investigated changes in retinal blood vessels across various age groups using light and transmission electron microscopy, TUNEL, and immunohistochemistry [92]. The analysis encompassed vascular smooth muscle cells, oxidative stress, microglia, and blood vessels to evaluate the extent of human retinal aging. The results revealed that starting from the age of seventy, the capillary endothelium and pericytes of the human retina undergo intricate changes [92].

As retinal capillaries and pericytes age, there is a notable loss of organelles and cytoplasmic filaments, coupled with a gradual thickening of the basal layer of vascular endothelial cells and pericytes. Accumulation of lipofuscin and autophagic vacuoles becomes prominent in vascular smooth muscle cells [92]. These findings underscore substantial age-related transformations in retinal perivascular cells and vascular smooth muscle cells, potentially limiting energy supply to neurons and contributing to age-related intraretinal neuron loss [92].

Changes in retinal blood vessels can serve as indicators of systemic blood vessel alterations. A study investigating the relationship among retinal age, arterial stiffness index, and cardiovascular disease events found a significant correlation between retinal age and both the arterial stiffness index and the incidence of cardiovascular events [93]. Senescent cells accumulate in the retinas of diabetic retinopathy patients and during the peak of destructive neovascularization in mouse models of retinopathy [94].

Stress and aging contribute to the accumulation of blood vessels and aging neurons in the retina [95]. Prolonged exposure to chronic activation of stress signaling responses and senescent cells may expose neighboring cells to an enhanced pathological senescence-associated secretory phenotype, leading to tissue dysfunction, multiple chronic diseases, and age-related pathologies [95]. The intricate interplay of aging, hypertension, and diabetes exerts complex and crucial effects on the microvascular structure. A study indicated that retinal microvascular remodeling is specifically associated with hypertension, while retinal blood vessel growth correlates with aging and hyperglycemia [96].

Figure 4 illustrates the main characteristics described in the aged retinal endothelium and vascular smooth muscle cell layer.

#### 3.2.3. Molecular Insights into Retinal Vascular Aging: The Pivotal Role of Oxidative Stress

The premature decline in vascular function observed in patients with diabetes is attributed to the aging of endothelial cells [97]. Multiple pathways contribute to this phenomenon, encompassing structural and functional alterations post-cellular aging, heightened oxidative stress, diminished nitric oxide production by endothelial nitric oxide synthase, alterations in gene expression, and an increase in pro-atherogenic responses [98]. Senescent endothelial cells and senescence-associated β-galactosidase (SA-β-gal)-positive cells have been identified in the aorta of diabetic rats exposed to elevated glucose concentrations [99]. The aging of vascular endothelial cells constitutes a pivotal factor in causing vascular dysfunction within the body. Clinical manifestations of vascular aging involve increased arterial stiffness and systolic blood pressure [100]. Oxidative stress emerges as a primary cause of vascular endothelial cell dysfunction and subsequent damage, accelerating the aging process and DNA damage in these cells [101]. Specifically, disruptions in the physiological redox homeostasis associated with vascular aging events seem to play a central role in ocular diseases such as AMD and diabetic retinopathy [51]. Research underscores the significant role of mitochondrial deacetylase, known as sirtuins (SIRT), in the aging process induced by glucose-induced oxidative stress, revealing its involvement in the regulation of vascular endothelial cell aging through miRNA [102].

Anomalous expression and activation of molecular signals, including nitric oxide (NO), endothelium-derived hyperpolarizing factor (EDHF), and calcium (Ca^2+^), contribute to increased vascular endothelial permeability, impaired neovascularization, vascular repair, and reduced mitochondrial biosynthesis in aging and senescent vascular endothelial cells [103]. Pathophysiological changes associated with vascular dysfunction involve cell cycle dysregulation, heightened reactive oxygen species (ROS), alterations in Ca^2+^ signaling, hyperuricemia, and vascular inflammation [103]. Proteins and signaling pathways such as Sirt1, Klotho, and FGF21, among others, play crucial roles in these pathophysiological changes. Furthermore, the accumulation of genetic damage and epigenetic changes adversely affect normal gene expression and activity, leading to cellular senescence and vascular dysfunction [103].

The dysregulation of angiogenesis, compounded by cellular senescence and persistent imbalances in nutrient and oxygen supply, triggers a stress response in retinal blood vessels [104]. Senescent endothelial cells, characterized by limited replicative potential and increased vulnerability to pathological assault, further contribute to this vascular challenge [105]. Research by Venkatesh et al. has shown that endothelial cell senescence is associated with heightened permeability, attributed to alterations in the expression and localization of vascular endothelial (VE)-cadherin and β-catenin [106].

The aging process of vascular endothelial cells can be induced through various mechanisms, with telomere shortening and damage to DNA and other cellular components implicated in the aging process [103]. Numerous molecular signals and signaling pathways play crucial roles in intravascular cell aging. Notably, SIRT1 and SIRT6 have been identified as preventive factors against vascular endothelial cell aging [107,108]. Conversely, Klotho and NRF2 act to prevent vascular endothelial cell senescence, while factors such as angiotensin, IGFBP3, IGFBP5, and mechanistic target of rapamycin (mTOR) promote senescence. For instance, in Ang II-induced aging, the pharmacological inhibitor of Ang II, olmesartan, has been found to inhibit vascular inflammation and premature aging [108]. Cellular senescence is closely related to SIRTs, with all SIRTs, except SIRT5, being expressed in the human retina. The loss of nicotinamide phosphoribosyltransferase (NAMPT) in aging RPE cells reduces the availability of NAD⁺ and the expression of SIRT1, promoting cell senescence [109,110].

By 2030, 20% of the world’s population will be over 65 years old, making age-related endothelial dysfunction a key risk factor for cardiovascular diseases. Vascular aging and its complications, including renal disease, neurodegenerative diseases, hypertension, and cancer, are increasingly recognized as “channelopathies” [111]. Notably, Ca^2+^ signaling in vascular smooth muscle plays a crucial role in arterial tone regulation and vascular function. During aging, changes in vascular structure and remodeling processes alter several conduction modes of Ca^2+^ signals, impacting vascular function [112].

Clinical experimental results indicate that the mechanisms driving vascular and cardiac aging phenotypes differ between men and women, emphasizing the need for gender-specific therapies to address the aging process and treat cardiovascular disease in the aging population [113]. Notably, testosterone exhibits anti-inflammatory properties, and optimal levels of androgens have been associated with improved longevity. However, during aging, there is a significant decline in these hormones in males, suggesting a correlation between aging, inflammation, and androgen levels [114]. One possible explanation for the age-related decline in androgens may be attributed to the increased synthesis of serum sex hormone-binding globulin, as demonstrated in a recent dedicated study [115].

Recent research has underscored the impact of traffic noise and air pollution exposure on cardiovascular aging and neurological disease. Traffic noise exposure, in particular, may lead to alterations in stress hormone levels, heart rate, and neuroinflammation. On the other hand, air pollution can trigger immune reactions and ROS formation through ultrafine particles, collectively inducing mitochondrial dysfunction, oxidative stress, telomere shortening, and chronic inflammation—key etiopathogenetic factors in cardiovascular and neurological disorders [116,117].

Moreover, UV irradiation plays a significant role in skin aging and ocular aging, serving as a recognized risk factor for cataracts and AMD [118,119,120,121]. In the context of AMD, UV exposure may accelerate retinal vascular aging by inducing ROS formation, triggering inflammatory responses, promoting neovascularization, and increasing the expression of MMP-2 and MMP-9, which degrade the extracellular matrix, thereby facilitating pathological neoangiogenesis [122].

High myopia has been identified as a significant pathophysiological trigger in various retinal disorders, further exacerbated by aging processes [123,124]. Eyes with high myopia often exhibit aberrations in the retinal microvascular network, characterized by reduced vessel angle, fractal dimension, vessel density, and vascular branches. These characteristics are closely associated with the severity of myopic maculopathy, axial elongation, best corrected visual acuity, and age [125]. It has been proposed that the connection between high myopia and vascular aging lies in the aggravating effects of age-related phenomena, such as decreased tissue perfusion and loss of endothelial cells, on the chorioretinal microvasculature, particularly impacting the diminished ocular blood flow observed in high myopia [68,82,126,127]. Consequently, these altered vascular conditions during aging may increase the susceptibility of high myopic eyes to vascular and age-related eye disorders, potentially leading to chorioretinal atrophy in high myopia [128].

Mitochondrial components activate inflammation, and aging influences blood–brain barrier function by affecting cerebrovascular mitochondrial function and inflammatory signaling [129]. Aging within the cerebrovascular results in the upregulation of interferon gene stimulators and an increase in interleukin-6 (IL-6), a cytokine that alters mitochondrial function. These changes may compromise the integrity of the blood–brain barrier and reduce cerebrovascular health with age [130].

Glaucoma, also an age-related neurodegenerative eye disease, exhibits a steadily increasing prevalence with advancing age [131]. The condition is characterized by the accelerated loss of retinal neurons and their axons, often coexisting with other age-related eye diseases that share molecular mechanisms resulting from repeated light damage and subsequent oxidative stress [132]. Aging contributes to abnormal peripheral blood flow, vascular stenosis, and endothelial dysfunction, all closely linked to the development of glaucoma [133]. Studies have found that in addition to increased intraocular pressure, an important risk factor, increased glutamate levels, changes in nitric oxide (NO) metabolism, vascular changes, and oxidative damage caused by ROS are also important factors in glaucoma [134]. Cellular senescence has been shown to lead to impaired angiogenesis, with multiple studies finding reduced retinal blood flow and capillary density in patients with glaucoma [133]. In patients with glaucoma with impaired retinal vascular function and reduced retinal vascular reactivity, RGCs and their axons are exposed to increased oxidative stress, which leads to further progression of glaucomatous damage [135]. A model of rat retinal ischemia–reperfusion injury demonstrated that oxidative stress triggers vascular senescence, which reveals viability through the upregulation of senescence-associated beta-galactosidase (SA-β-gal) activity, senescence-related protein p53 and p16 activity, and a reduction in retinal ganglion cells [136].

Aging is also closely related to diseases related to retinal vascular occlusion, and the risk of branched retinal vascular obstruction (BRVO) and central retinal vein occlusion (CRVO) increases with age and concurrent cardiovascular disease [137]. Retinal vascular obstruction (RVO) mainly occurs among the elderly, with more than 50% of cases occurring in people over 65 years [138]. The pathogenesis of RVO is not fully understood, but multiple molecules have been found to be related to the occurrence of RVO [76]. For example, IL-6 is up-regulated in RVO, especially in CRVO [139]. In mice, aging is associated with increased levels of the inflammatory cytokine IL-6 in the aorta, and IL-6 participates in a positive feedback loop with impaired vascular mitochondrial function, promoting the progression of aging [140]. IL-8 has also been found to play an important role in BRVO and ischemic RVO. IL-6 and IL-8 are consistently present in the senescence-associated secretory phenotype (SASP) [141]. MMP-9 has also been found to be closely related to RVO, and it is an important molecule involved in the aging process [142,143]. Table 1 outlines the key pathophysiological drivers and their associated molecular pathways that play pivotal roles in vascular aging processes.

## 4. Emerging Therapeutic Strategies and Novel Targeting Pathways

Following extensive research, molecular markers of vascular aging have been continuously identified, encompassing increased oxidative and nitrative stress responses, impaired oxidative stress resistance, chronic low-grade sterile inflammation via an NF-κB activation, cytokine dysregulation, damage-associated molecular patterns (DAMPs), mitochondrial dysfunction, NAD+ depletion, SIRT1 dysregulation, mTOR dysregulation, AMP-activated protein kinase (AMPK) dysregulation, ΔDNA methylation, miRNA dysregulation, loss of proteostasis, progenitor cell depletion, endogenous DNA damage (activation of p53-p21 and p16INK4a-pRb pathways), and mitochondrial DNA alterations [27,144]. This intricate process involves a decrease in mtDNA copy number and an increase in mtDNA mutations in various tissues of aging organisms. Additionally, oxidative stress triggers the release of mtDNA into the cytoplasm, where it binds to cGAS and contributes to cellular senescence by activating the cyclic GMP-AMP receptor stimulator of interferon genes (STING) [46,144].

### 4.1. Modulation of the eNOS/NO Pathway

Oxidative stress impacts vascular function through protein oxidation, redox-sensitive transcription factor induction, and, notably, the deactivation of endothelium-derived nitric oxide (NO) [145]. Factors influencing NO bioavailability encompass ROS, endothelial NO synthase (eNOS) activation status, and changes in substrate (L-arginine) and cofactor (tetrahydrobiopterin, BH_4_) availability [27]. Increased endothelin 1 and/or reduced eNOS expression may also lead to age-related changes, ultimately reducing NO bioavailability [146]. Oxidative stress activates redox-sensitive cell signaling pathways, including NF-kB, implicated in the inflammatory process of aging vasculature. Activation of matrix metalloproteinases (MMPs) further contributes to vascular structure destruction [27]. The NO-mediated guanylate cyclase/cGMP pathway, critical for ocular blood flow and neuroprotective effects, also plays a significant role [147]. The eNOS/NO pathway improves vascular health, partly through crosstalk with HO-1/CO [148]. BH_4_ is an important eNOS cofactor required for efficient electron transfer in the eNOS catalytic cycle and largely determines its activity [149]. Given the critical role of BH_4_ in eNOS activity and endothelial health, BH_4_ supplementation interventions to improve vascular function are a worthy research direction. The efficacy of BH_4_ treatment in preventing eNOS uncoupling has been demonstrated in animal models as well as human studies [150]. Because BH_4_ is very unstable and easily oxidized to pterin, it has been suggested that the co-administration of BH_4_ with antioxidants may be an idea for using BH_4_ supplements [151]. But further research is needed on the relationship between BH_4_ supplements and the treatment of vascular aging.

Research indicates that oxidative stress reducers may have anti-aging effects on the retina. In endothelial cells, maintaining appropriate levels of VEGF and endothelial nitric oxide synthase (eNOS)/NO is crucial for endothelial cell survival [152]. Drugs such as Rapamycin, TNF-α antibodies, and therapeutic NADPH oxidase inhibitors or antioxidant compounds improve NO bioavailability by increasing production and/or reducing NO degradation caused by oxidative stress [153].

### 4.2. Nrf2 Activators and Mitochondria-Targeting Molecules

The redox-sensitive transcription factor nuclear factor erythroid 2–related factor 2 (Nrf2), crucial in cellular antioxidant defense and redox homeostasis, emerges as an attractive drug target for anti-aging interventions [153]. Another transcription factor, the nuclear factor kappa-B (NF-kB) is an important transcription factor expressed in all mammalian cells and regulates gene expression of factors such as cell proliferation, adhesion, inflammation, redox status, and tissue-specific enzymes [154]. As a major regulator of the SASP, NF-kB is composed of proteases (matrix metalloproteinases), inflammatory cytokines (IL-6 and IL-8), chemokines (monocyte chemoattractant protein and Macrophage inflammatory protein), and growth factors (granulocyte-macrophage colony-stimulating factor and transforming growth factor-β) together play a significant role in the aging process [155]. Increased ROS production and exacerbated Nrf2 dysfunction enhance NF-κB activation, promoting inflammatory cytokine and chemokine expression, microvascular endothelial activation, leukocyte adhesion, and extravasation. Elevated nitro-oxidative stress further promotes PARP1 activation, leading to impaired activity of anti-inflammatory sirtuin proteins [27]. These molecular mechanisms of aging present positive avenues for treating age-related diseases, with pharmacological treatment being the primary strategy against aging. Anti-aging drugs work by reducing the number of senescent cells, alleviating SASP, exerting anti-inflammatory and antioxidant effects, and influencing multiple signaling pathways simultaneously [144].

Modulating Nrf2, particularly, activating it, emerges as a promising strategy to mitigate aging processes and associated oxidative stress [156]. In this regard, α-lipoic acid stands out as a promising molecule. Studies have shown its ability to alleviate oxidative stress in the retina by upregulating heme oxygenase-1 (HO-1) expression through activation of the Keap1/Nrf2 signaling pathway [157,158,159].

In the aging vasculature, increased mitochondrial reactive oxygen species (mtROS) due to mitochondrial damage contributes to electron transport chain dysfunction. This process can be triggered by peroxynitrite-mediated nitration, MnSOD inhibition, and decreased cellular glutathione content. Down-regulation and/or damage of p66 Shc exacerbates the Nrf2-mediated antioxidant defense response [27]. Importantly, mtROS stands out as a pharmacological target for vascular protection. The use of the mitochondrial antioxidant MitoQ effectively reverses age-related endothelial dysfunction and restores NO bioavailability [160].

Resveratrol, a potent antioxidant, significantly attenuates mtROS production in endothelial and smooth muscle cells [161]. Treatment with oxidized tetrapeptide SS-31 has also demonstrated improvement in arterial endothelial function in rodent models of aging [76]. Mitochondria-derived H2O2 has been linked to low-grade vascular inflammation during aging, inducing NF-κB activation in endothelial cells and smooth muscle cells [162]. Another crucial link between mtROS production and vascular aging is the induction of apoptosis through the Bcl-2-dependent pathway [163].

### 4.3. SIRT Activators, Telomerase Enhancers, and Senolytics

Pharmacological inhibition of the PARP pathway emerges as a potentially significant therapeutic target for aging and age-related diseases. PARP-1 activation upregulates NF-κB-dependent inflammatory gene expression, closely associated with aging. PARP-1, an NAD+-consuming enzyme, competes with SIRT1 for the same NAD+ pool. Increased PARP-1 activity leads to SIRT1 inhibition, making PARP-1 inhibitors protective against aging vessels [153,164].

SIRT1, with its potent anti-inflammatory effect, plays a crucial role in preventing vascular inflammation during aging. Pharmacological activators of SIRT1, such as resveratrol and SRT1720, have demonstrated a reduction in vascular inflammation in aged mice [153,165,166]. Eliminating senescent cells expressing p16INK4A has been shown to extend the lifespan and health span of mice [13]. Cycloastraganol, the main natural compound in Astragalus membranaceus, exhibits various pharmacological effects, including anti-aging, anti-inflammatory, and anti-fibrotic effects [167]. Telomerase activity, which is highest in human embryonic tissue, gradually decreases with age [165]. The intervention role of telomerase in aging-related diseases has been confirmed [168]. In this context, Cycloastraganol represents the only discovered telomerase activator so far, being able to delay telomere shortening by inducing telomerase expression [169]. Senolytics, designed to identify and target senescent cells, include common drugs like dasatinib and quercetin [170,171]. Bcl-2 inhibitory drugs UBX1325 and UBX1967, treating AMD, promote apoptosis in senescent cells [172]. Sulodexide has a delaying effect on senescent cells, preventing cellular senescence in cultured endothelial cells and slowing down hyperglycemia-dependent senescence, reducing the angiogenic and inflammatory effects of these cells [88,173].

### 4.4. Antidiabetics as Anti-Aging Drugs

Experimental evidence indicates that dapagliflozin can prevent impaired endothelium-dependent vasodilation and enhance arterial stiffness in type 2 diabetes (T2D) mouse models [174]. Moreover, dapagliflozin reduced the aging activity of the aorta and levels of aging-related inflammatory factors in the experimental mouse group [174]. The compound demonstrates efficacy in improving senescence activity, protein markers of senescence, and oxidative stress during in vitro cell culture [175]. Additionally, dapagliflozin has been shown to mitigate the decline in endothelial nitric oxide synthase (eNOS) phosphorylation and nitric oxide production in aging vascular endothelial cells [175]. Notably, dapagliflozin enhances the expression of mitochondrial deacetylase 1 (SIRT1) in high glucose-induced aging vascular endothelial cells [175]. The anti-aging effect of dapagliflozin is dependent on SIRT1 activation, highlighting the pivotal role of SIRT1 signaling in preventing endothelial cell senescence in diabetic mice [175].

Figure 5 presents a schematic summary of the main pathophysiological drivers in retinal vascular aging and the proposed emerging novel treatment targets to contrast mitochondrial dysfunction, oxidative stress, and NO alterations.

### 4.5. Complement Inhibitors

As research and science progress, significant strides have been made in treating common eye conditions associated with aging. For example, anti-vascular endothelial growth factor (anti-VEGF) drugs have revolutionized the treatment of AMD [65,176]. The continuous discovery of aging-related molecular mechanisms provides numerous potential targets and research directions for the treatment of age-related eye diseases. As the connection between the complement system and AMD becomes increasingly clear, many innovative therapeutic attempts targeting complement components have emerged in the treatment of AMD. For example, pegcetacoplan, a specific inhibitor of complement protein C3, has been found to inhibit geographic atrophy (GA) that occurs in AMD [177]. Zimura, also known as avacincaptad pegol, a PEGylated RNA aptamer, is a specific and potent C5 inhibitor that has been proven to be effective in slowing down GA in the first two phases of clinical trials, but a phase III trial is still needed [178]. There are also many complement inhibitors still in the clinical trial stage, such as NGM621, AAVCAGsCD59, and IBI302 [179].

Table 2 outlines the primary emerging molecular targets that hold the potential for mitigating ocular vascular aging.

## 5. Conclusions

Vascular aging stands as a pivotal intermediary between the aging process itself and the onset of age-related diseases. The aging vasculature undergoes a multitude of biochemical alterations and structural remodeling. The rate at which vascular aging unfolds is influenced by specific risk factors, as well as genetic and epigenetic components. Within the field of ophthalmology, particular attention has been devoted to the aging of choroidal and retinal blood vessels. Numerous factors have been identified that directly and indirectly inflict damage on the endothelium, instigating changes associated with vascular aging. Common pathways are shared between vascular aging in various tissues and ocular structures. This review underscores the multifaceted nature of influences on vascular aging. Hyperglycemia, arterial hypertension, oxidative stress, mitochondrial dysfunction, telomere attrition, alterations in sex hormone levels, exposure to traffic noise and air pollution, UV irradiation, and high myopia collectively contribute to the progression of aging-related phenomena. Simultaneously, at the molecular level, an intricate interplay of factors and signaling pathways is implicated in the aging process. Through this comprehensive exploration, our objective is to unveil potential mechanisms and therapeutic strategies for addressing diseases intricately linked to vascular aging.

## Figures and Tables

**Figure 1 biomedicines-12-00817-f001:**
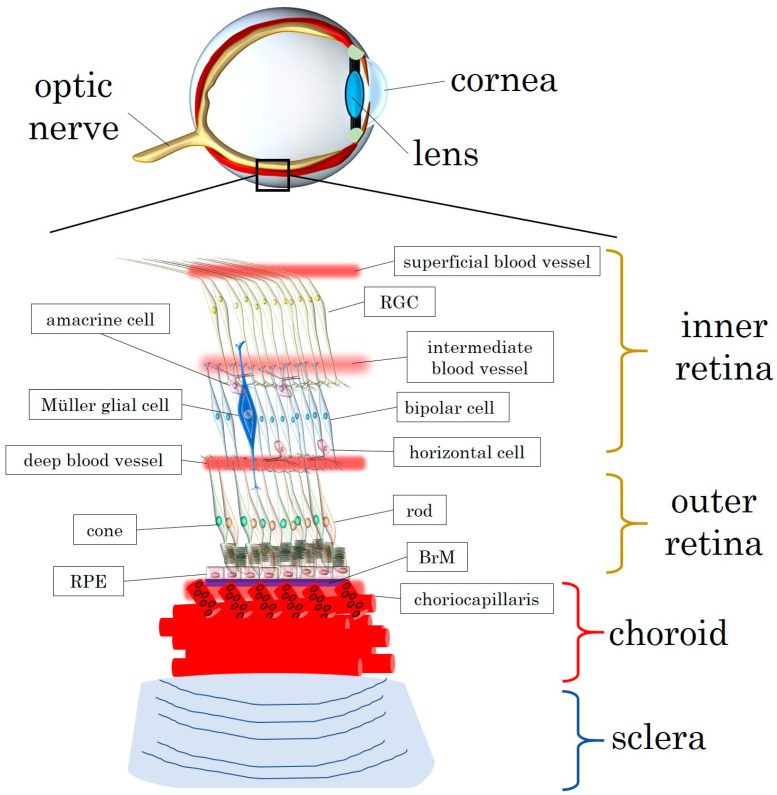
Illustration of the retinal layers and the structure of the choroid. BrM: Bruch’s membrane; RGC: retinal ganglion cell; RPE: retinal pigment epithelium.

**Figure 2 biomedicines-12-00817-f002:**
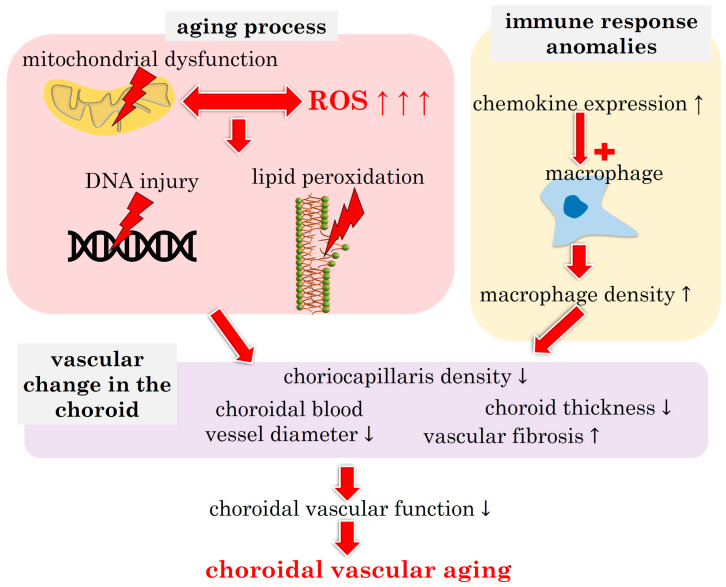
Schematic presentation of the pathophysiological drivers in choroidal vascular aging. ROS: reactive oxygen species. Upward arrows indicate increase, downward arrows indicate decrease.

**Figure 3 biomedicines-12-00817-f003:**
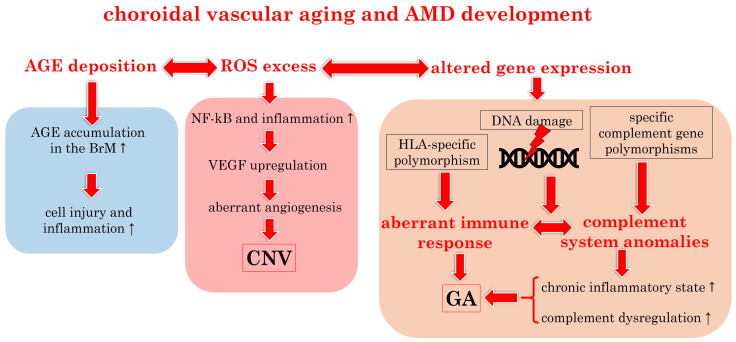
Pathophysiological link between choroidal vascular aging and the occurrence of AMD. AGE: advanced glycation end product; AMD: age-related macular degeneration; BrM: Bruch’s membrane; CNV: choroidal neovascularization; GA: geographic atrophy; HLA: human leucocyte antigen; NF-kB: nuclear factor “kappa-light-chain-enhancer” of activated B-cells; ROS: reactive oxygen species; VEGF: vascular endothelial growth factor. Upward arrows indicate increase.

**Figure 4 biomedicines-12-00817-f004:**
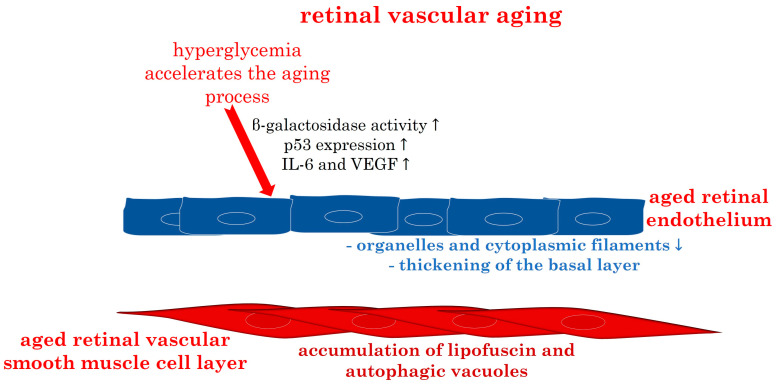
Overview of the reported characteristics in the aged retinal endothelium and vascular smooth muscle cell layer. IL: interleukin; VEGF: vascular endothelial growth factor. Upward arrows indicate increase, downward arrows indicate decrease.

**Figure 5 biomedicines-12-00817-f005:**
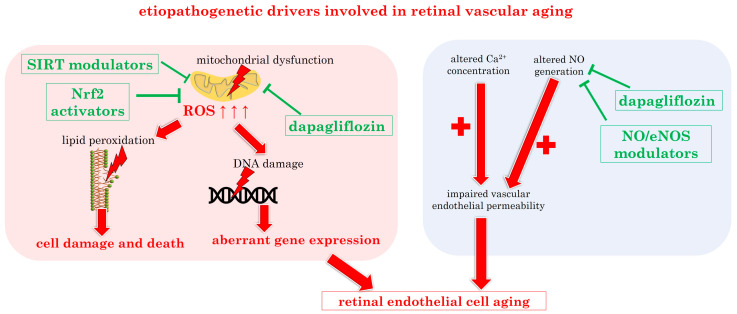
Overview of the reported pathophysiological drivers in retinal vascular aging and the novel treatment targets. eNOS: endothelial nitric oxide synthase; NO: nitric oxide; Nrf2: nuclear factor erythroid 2-related factor 2; ROS: reactive oxygen species; SIRT: sirtuin. Upward arrows indicate increase.

**Table 1 biomedicines-12-00817-t001:** Summarizing table proposing the main pathophysiological triggers and mechanisms involved in vascular aging.

Pathophysiological Trigger	Pathomechanism	Refs.
Hyperglycemia	retinal vascular growth and aging, with increased β-galactosidase activity, p53 expression, elevated IL-6 and VEGF	[88,89,90,96]
Hypertension	retinal microvascular remodeling	[96]
Oxidative stress	pivotal factor in vascular endothelial cell dysfunction and subsequent damage, exacerbating aging process, cell death and DNA damage in several ocular disorders	[51,101]
Mitochondrial dysfunction	reduced mitochondrial biogenesis and SIRT activity	[102,103,129]
Telomere alteration	telomere shortening and damage to DNA	[17,103]
Ca^2+^ signaling anomalies	remodeling processes induce impairment in Ca^2+^ signals, impacting vascular function	[112]
Angiogenetic dysregulation, endothelial dysfunction	accumulation of genetic damage and epigenetic changes affecting gene expression, leads to vascular impairment	[103,104,105,133,137,138]
Vascular obstruction	aging may exacerbate the pro-inflammatory activity of IL-6, IL-8, and MMP-9, all agents related to retinal vascular occlusion	[139,140,141,142,143]
Androgen decline	enhanced synthesis of the serum sex hormone-binding globulin may lead to a reduced level of androgens, which physiologically exert an anti-inflammatory effect	[114,115]
Traffic noise and air pollution exposure	-Traffic noise exposure: alteration in stress hormone levels and heart rate as well as establishment of neuroinflammation. -Air pollution: ultrafine particles induce immune reactions and ROS formation.collectively, these factors trigger mitochondrial dysfunction, oxidative stress, telomere shortening, and chronic inflammation	[116,117]
UV exposure	raised ROS formation, and subsequently elevated VEGF, MMP-2, and MMP-9, causing neoangiogenesis and degradation of ECM	[122]
High myopia	age-related decreased tissue perfusion and loss of endothelial cells, aggravating the reduced ocular blood flow in high myopia	[68,82,126,127]

**Table 2 biomedicines-12-00817-t002:** Overview of promising agents for combating oxidative stress and processes of vascular aging.

Mechanistic Target	Candidate Molecule	Refs.
eNOS modulation	BH_4_	[150,151]
Mitochondria	MitoQ	[160]
	tetrapeptide SS-31	[76]
Nrf2 activation	α-lipoic acid	[156,157,158,159]
SIRT1 activation	PARP-1 inhibitors	[164]
	resveratrol	[166]
	SRT1720	[165]
	dapagliflozin	[174,175]
mTOR inhibition	rapamycin	[180,181]
Telomerase activation	cycloastraganol	[169]
Senolytic	dasatinib	[170]
	quercetin	[170]
	Bcl-2 inhibitors	[172]
	Sulodexide	[88,173]
Complement inhibition	pegcetacoplan	[177]
	avacincaptad pegol	[178]
	NGM621, AAVCAGsCD59, IBI302	[179]

## Data Availability

Not applicable.

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
