# Peer review of "Aging in Ocular Blood Vessels: Molecular Insights and the Role of Oxidative Stress"

_biomedicines, 2024, doi:10.3390/biomedicines12040817_

Round 1

Reviewer 1 Report

Comments and Suggestions for Authors

The review work, “Aging in Ocular Blood Vessels: Molecular Insights and Role of Oxidative Stress” by Cui et al. focusing on the impact of vascular aging on ocular 13 blood vessels and related diseases along with illuminate molecular insights contributing to vascular cell aging, with a particular emphasis on the choroid and retina. The work summarized more than 150 literatures in this field with excellent illustration in this field. However, summarizing data in tabular form is lacking along with comparison of work done with each other.  The specific comments are as follows:

1.       Line 49: Please add preface after 2. The Concepts of Aging and Vascular Aging before starting 2.1

2.       Line 60: Current research, please rewrite as work already done of ref 15.

3.       Line 111: Please add preface after 3. Vascular Aging in Ocular Blood Vessels.

4.       Line 246-248: Interestingly, an opposite pattern of CFH was observed in early-stage and wet AMD eyes, with significantly lower CFH levels in the choriocapillaris and BrM compared with age-matched controls. Please verify the sentence and possible give more supporting literature to this.

5.       Figure 3 summarizes the connection between choroidal aging and the development of AMD, if the connections are any literature please cite this too.

6.       Strongly recommend to add two to three summarized tabular data in this filed for better understanding of reader.

7.       Concluding remarks need to write as conclusion.

Author Response

We thank the reviewer for the comments and suggestions. According to these suggestions, we made changes in the text.

Specific comments:

  1. Line 49: Please add preface after 2. The Concepts of Aging and Vascular Aging before starting 2.1

Response to 1.): We added a passage accordingly (lines 49–53, new passages are underlined).

  1. Line 60: Current research, please rewrite as work already done of ref 15.

Response to 2.): We modified it accordingly (lines 66–69, changes are underlined).

  1. Line 111: Please add preface after 3. Vascular Aging in Ocular Blood Vessels.

Response to 3.): We added a section accordingly (lines 119–122, the new section is underlined).

  1. Line 246-248: Interestingly, an opposite pattern of CFH was observed in early-stage and wet AMD eyes, with significantly lower CFH levels in the choriocapillaris and BrM compared with age-matched controls. Please verify the sentence and possible give more supporting literature to this.

Response to 4.): We added a reference and more details in the text accordingly (lines 252–257, changes are underlined).

  1. Figure 3 summarizes the connection between choroidal aging and the development of AMD, if the connections are any literature please cite this too.

Response to 5.): We added a reference accordingly (lines 270–271, changes are underlined).

  1. Strongly recommend to add two to three summarized tabular data in this filed for better understanding of reader.

Response to 6.): We added accordingly a summarizing table (table 1) at the end of chapter 3, and another summarizing table (table 2) at the end of chapter 4.

  1. Concluding remarks need to write as conclusion

Response to 5.): We modified it accordingly.

Reviewer 2 Report

Comments and Suggestions for Authors

This review article demonstrates multiple factors exert such as hyperglycemia, oxidative stress, mitochondrial damage, telomere damage, changes in sex hormone levels, and exposure to air pollution influence on the progression of vascular aging, and reveals some factors and signaling pathways involved in the vascular aging. It is well-written and provides clinical significance, however the authors would be better to investigate the effect of UV exposure and high myopia (important risk factors of retinal diseases) on the vascular aging.

Author Response

We thank the reviewer for the comments and suggestions. According to these suggestions, we add supplements in the text.

  1. Effect of UV exposure on the vascular aging

Response to 1.): We added a passage accordingly (lines 430–435, new text passages are underlined).

  1. Effect of high myopia on vascular aging

Response to 2.): We added a passage accordingly (lines 436–447, new text passages are underlined).